# Compost Quality and Sanitation on Industrial Scale Composting of Municipal Solid Waste and Sewage Sludge

**Ana B. Siles-Castellano, Juan A. López-González \*, Macarena M. Jurado, María J. Estrella-González, Francisca Suárez-Estrella and María J. López**

Unit of Microbiology, Department of Biology and Geology, CITE II-B, Agrifood Campus of International Excellence ceiA3, CIAIMBITAL, University of Almeria, 04120 Almeria, Spain; asc426@ual.es (A.B.S.-C.); mjr956@ual.es (M.M.J.); meg274@ual.es (M.J.E.-G.); fsuarez@ual.es (F.S.-E.); mllopez@ual.es (M.J.L.)
\* Correspondence: lgj132@ual.es; Tel.: +34-950-015-891

**Abstract:** Municipal solid waste and sewage sludge are produced in large quantities that are often managed through industrial composting treatment. Because of their origin, composition, and complexity, ensuring adequate stabilization of the organic matter, and sanitation of fecal contaminants during composting is of the utmost significance, and difficult to achieve on an industrial scale. In this study, the operations of six industrial composting facilities that process municipal solid waste and sewage sludge were evaluated from the point of view of the sanitation achieved and the quality of the compost produced. In addition, the results were compared using the model of industrial compost from green waste. Differences between the plants were ascribable to operations other than composting systems. High phytotoxicity and fecal contamination above legislation thresholds were found in compost produced from municipal solid waste. In contrast, compost from sewage sludge were more stable and mature than those produced from green waste, and also had an adequate level of sanitation. The raw material and operational factors are of great relevance to obtain a stable, mature, and pathogen-free compost.

**Keywords:** fecal contamination; *Salmonella*; fecal coliforms; maturation; stability; industrial compost



## 1. Introduction

At present, the amount of wastes generated by the welfare society has increased considerably, and the number of industrial facilities dedicated to organic waste treatment has enlarged notably [1]. Industrial composting has become the most widespread organic waste treatment procedure. There is a need for a tool that unites economic activity and environmental well-being in a sustainable way that also integrates organic waste into the circular economy model implemented in Europe [2]. This process results in the stabilization of organic matter, generating a final product that can be used as a soil conditioner and a supply of nutrients [3,4]. There is a need to take into account the particularities of industrial composting with respect to small-scale composting in order to avoid the risk of generating products that are unsuitable for agronomic application [5]. If not, the risk is high.

In a composting process, the starting raw materials can be very diverse in composition and origin [6]. The knowledge of the characteristics of the raw materials is essential to provide the optimal composting conditions to obtain good quality compost [7]. There are many studies in literature dealing with specific bio-materials and co-composting mixtures [7–9], among which urban waste, i.e., municipal solid waste (MSW) and sewage sludge (SS), are of special interest because of its problematic management [9]. Most MSW typically includes organic residues from households, markets and food processing that contribute to its high organic content and therefore biodegradable fraction. This fraction usually consists of more than 50% of the total waste generated and can be as high as 80% for source separated MSW [10,11]. Hence, MSW is a highly compostable waste [12]. The use of compost produced from MSW has increased in recent years. Its use exerts a positive effect

on the activities of enzymes involved in the carbon, nitrogen and phosphorus cycles [13] and provides a high amount of organic matter [14]. Therefore, it is useful as an agricultural soil conditioner and fertilizer [15]. In this sense, the other main waste material of urban origin, sewage sludge, has management alternatives that include even thermochemical decomposition [16]. Regarding the composting of SS, some key aspects must be taken into account. SS possess certain characteristics that are common to MSW, which makes it also suitable for composting. However, due to its low C/N ratio, requires co-composting with, e.g., lignocellulosic waste [17]. Among the most negative aspects of MSW and SS are the content of heavy metals and the presence of recalcitrant xenobiotic substances, which are capable of migrating and distributing through environments [18,19]. Besides, research is increasingly focused on the study of techniques to improve the quality of the final product obtained, without forgetting the conservation of natural resources [20] and the reduction in environmental impact [21]. For the compost to be suitable for agriculture, maturity and stability should be ensured. The stability is often evaluated using seed germination tests [7]. However, respirometry is one of the best tools to monitor the stability of the material during the composting process [22]. The maturity identifies the grade of organic matter transformation during the process and the validation of the elimination of phytotoxicity in the compost generated is preferred [23]. On the other hand, a correctly prepared compost must not contain pathogens that pose risks to health and the environment. Fecal contamination indicators are used to determine that the compost is safe [24] and to verify that the sanitation conditions were reached during the process. This is especially important for large pile sizes used at industrial scale composting in which it is difficult to establish that the desired conditions are reached in the whole mass.

Due to the above considerations, it is important to conduct a study to evaluate whether the urban waste bio-materials and the conditions of the composting process influence the quality of the compost obtained. In this study, a comparative analysis is carried out of the compost produced from industrial composting facilities that process municipal solid waste (MSW) and sewage sludge (SS) and compare them with those produced from green waste (VR). To achieve this goal, the following specific objectives were realized: (1) to evaluate the physico-chemical characteristics of raw materials in different industrial facilities processing urban and green waste; (2) to determine the evolution of the organic fractions of these bio-materials during composting; (3) to estimate the presence of fecal contamination in the final compost as an indicator of sanitization in industrial composting processes; (4) to compare the composts generated in the processes, taking into account physico-chemical properties, biological stability, maturity and fecal contamination of the substrates.

## 2. Materials and Methods

### 2.1. Sampling Strategy and Raw Materials

This study was carried out in different industrial composting companies that process different raw materials, located in the southeast of Spain (Almeria, Granada, Murcia and Alicante). Three facilities dedicated to each of the following organic waste were selected: vegetal residue (VR), municipal solid waste (MSW) and sewage sludge (SS). Table 1 shows the main treatment characteristics of each composting process in each facility. The composting piles were prepared outside, in a row of 7–15 m in length and 3–5 m. The samples were taken during the raw material (RM) and final product phases (FP). During RM and FP phases, samples corresponding to nine different points of the composting pile were taken. Three samples were taken superficially (0.5 m), three were taken at a depth of 1.5 m and the last three as close as possible to the bottom of the piles. For this, a probe that reached 1–2 m deep was used. Each of the samples obtained from the different points was mixed to obtain a representative sample, obtaining a final mass of 3 kg, which was divided into three sub-samples of about 1 kg. After, the samples were stored in vacuum bags and frozen at −20 °C, for further analysis.

**Table 1.** Characteristics of the industrial composting processes.

| Facility [a] | Waste Mixture | Method of Composting | Time of Composting Process (Months) |
|---|---|---|---|
| VR1 | Cucumber and zucchini crop residues: stalks, leaves | Open air, turned wind-rows | 4 |
| VR2 | Cucumber and zucchini crop residues: stalks, leaves | Open air, turned wind-rows | 4 |
| VR3 | Pepper crop residues: stalks, leaves | Open air, turned wind-rows | 3 |
| MSW1 | Municipal solid waste [b] | In-vessel turned wind-rows in bays | 3.5 |
| MSW2 | Municipal solid waste [b] | In-vessel turned wind-rows in bays | 4.5 |
| MSW3 | Municipal solid waste [b] | In-vessel tunnel composting (turning by augers) | 3 |
| SS1 | Sewage sludge + straw 1:1 *v/v* | Open air, turned wind-rows | 3.5 |
| SS2 | Sewage sludge + pruning wastes 1:1 *v/v* | Open air, turned wind-rows | 3 |
| SS3 | Dried sewage sludge + pruning wastes 1:2 *v/v* | In-vessel tunnel composting (turning by augers) | 3 |

[a] Vegetal residue (VR): pruning and gardening remains, agricultural (horticultural) or from agriculture; sewage sludge (biosolids) (SS): crude, active, sludge; municipal solid waste (MSW): organic fraction of municipal solid waste. [b] All facilities processed mixed municipal solid waste.

### 2.2. Analytical Methods

#### 2.2.1. Physico-Chemical Parameters

Different physico-chemical analyses were carried out on the bio-materials and industrial composts produced. Moisture was determined at 105 °C for 24 h. The electrical conductivity (EC) and pH were measured by preparing a 1/10 (*w/v*) extract in water. The organic matter content (OM) was determined by introducing a muffle furnace at 550 °C for 3.5 h. The determination of total carbon (C) and total nitrogen (N) was carried out using an Elementar Vario Micro CHNS (Elementar Analysensysteme GmbH, Hanau, Germany).

Soluble organic carbon (SOC), reducing sugars (RS) and total proteins (TP) were analyzed using the described methods [25], a brief description of these methods is provided next. For SOC a 1/10 dilution of the samples was performed in 0.5 M $K_2SO_4$ and shaken for 30 min at 200 rpm. Subsequently, filtered extracts were obtained from this dilution and measured using a TOC-VCSN analyzer (Shimadzu, Co., Kyoto, Japan). Reducing sugars (RS) were analyzed by the DNS method [26]. Total proteins (TP) were spectrophotometrically analyzed according to the method described by Lowry and modified by [27]. For $N-NH_4^+$ and $N-NO_3^-$, a 1/10 dilution was made and shaken for 30 min at 200 rpm. For the determination of $N-NH_4^+$, a Hach 9663 probe (Hach, Loveland, CO, USA) was used. For $N-NO_3^-$ Nitrachek 404 probe (KPG Products Ldt., Hove, UK) was used.

The cellulose (CEL), hemicellulose (HC) and lignin (LIG) fractions were determined using the ANKOM 200/220 analyzer (Ankom Technology, Macedonia, NY, USA). First, samples were subjected to digestion by using a neutral detergent (NDF), obtaining hemicellulose, cellulose and lignin fractions. After digestion using an acid detergent (ADF), the sum fraction of cellulose and lignin was obtained. Finally, this sum fraction was subjected to a treatment with concentrated sulfuric acid (ADL), the fraction that was obtained contained lignin (http://www.ankom.com/procedures.aspx (accessed on 10 June 2021)).

#### 2.2.2. Indicator Parameters of Biological Stability and Maturity

The phytotoxicity of the samples was evaluated, following the method of [28], and slightly modified and according to [7]. The biodegradability of the samples was evaluated using the dynamic respiration index (DRI) [29] and dynamic accumulated respiration activity ($AT_4$) [5]. For this purpose, 100 g of the sample, with 60% humidity, were placed in a 500 mL reactor and placed in a water bath at 37 °C. An oxygen sensor (Alphasense Ltd., Essex, UK) was used to measure the exhaust air. The values of the oxygen levels were controlled by a data acquisition system. For the DRI calculation, the average value

of oxygen consumed in the most active 24 h of biological activity was used, that is, as g of oxygen consumed per kg of organic matter (OM) and per hour (g O2 kg$^{-1}$ OM h$^{-1}$). AT$_4$ was expressed as g of oxygen per kg of OM matter (g O2 kg$^{-1}$ OM) after four days of cumulative respirometric activity.

### 2.2.3. Fecal Contamination and Pathogens

For the determination of each of the indicator groups of fecal contamination, 10 g of the different composts samples were suspended in 90 mL of sterile saline (NaCl, 0.9%) and stirred for 30 min at 120 rpm. Serial decimal dilutions of the samples were performed in an appropriate medium for each microbial group. The determination of total and fecal coliforms, *Escherichia coli* and fecal enterococci was carried out using the most probable number (MPN) technique by inoculating 1 mL of the dilutions made in each specific medium. For the quantification of total coliforms, 10 mL tubes of lactose broth (Sharlab, S.L., Barcelona, Spain) with 0.2 g L$^{-1}$ bromocresol purple and Durham bell, were used as a culture medium and it were incubated at 37 °C for 24–48 h in a water bath. After the incubation period, the total coliform count was carried out, as indicated by the change in the color of the medium from purple to yellow and the accumulation of gas in the Durham bell. The determination of fecal coliforms was made from positive total coliforms. For this, 0.1 mL of the positive tubes were taken and transferred to a tube with 10 mL of lactose broth with bromocresol purple and Durham bell which was incubated at 44.5 °C for 24–48 h. Again, as in the case of total coliforms, positives were searched for by the same method. From the positive fecal coliforms, the positive tubes were quantified by streaking with a platinum loop in eosin methylene blue agar (Panreac, ITW, Glenview, IL, USA) and incubating at 37 °C for 24–48 h. The appearance of colonies with a metallic luster indicated the presence of *E. coli*. The fecal enterococci were determined by tubes with 10 mL of Rothe broth (Oxoid Ldt., Hant, UK) and incubating at 37 °C for 24 h in a water bath. The presence of fecal enterococci was confirmed by the presence of turbidity and/or sedimentation in the tubes and confirmation by Gram staining and microscopy.

The quantification of the level of sulfate-reducing clostridia was performed by counting colonies per gram of sample (cfu g$^{-1}$) in SPS agar. For this, 1 mL of the different serial dilution was taken and transferred to a tube containing 15 mL of SPS agar (Sharlab, S.L., Barcelona, Spain), sealed with paraffin creating anaerobic conditions and incubated at 37 °C for 24–48 h. Colonies of sulfite-reducing clostridia showed a characteristic black color.

The detection of *Salmonella* spp. was carried out by weighing 25 g of the sample that was placed in a flask with 0.1% buffered peptone water (Panreac, ITW, Glenview, IL, USA) and incubating at 37 °C for 24 h. Subsequently, 1 mL was taken which was placed in a tube with 10 mL of selenite and cystine broth (Oxoid Ldt., Hant, UK). After the incubation time for the broth media, it was inoculated on Hektoen Enteric Agar (Panreac, ITW, Glenview, IL, USA) and incubated at 37 °C for 24 h. After the incubation time, the presence of *Salmonella* was detected by the appearance of green colonies with or without a blackened area. Suspected colonies were biochemically confirmed by inoculation in Kligler iron agar (Panreac, ITW, Glenview, IL, USA) and incubated at 37 °C for 24 h.

### 2.3. Data Analysis

The parameters analyzed were performed in triplicate, using the mean for the presentation of the data. The normality and homogeneity of the variances were verified using the Shapiro–Wilk and Levene tests. Statistical analysis was performed at a significance level of $p < 0.05$. By means of the analysis of variance (ANOVA) and the Fischer comparison test of least significant difference (LSD), the mean values for each sampling facility were compared. The presence of categories within final compost samples collected from facilities composting different raw materials was investigated using stepwise linear discriminant analysis (DA), in order to find simple equations for estimation of the composition of these wastes from easily analyzable parameters. Spearman correlations between different parameters and a multiple regression analysis with stepwise selection of variables were calculated.

Statgraphics Centurion (Version 18.1.8) (Stat-Point Inc., The Plains, VA, USA) was used for the analysis.

## 3. Results and Discussion

### 3.1. Bio-Materials Characterization

The physical-chemical state of the bio-materials that are composted is essential to guarantee an efficient process. Especially in industrial facilities, where the volume of waste treated can cause additional difficulties [5]. Table 2 shows the values recorded in raw materials in the physical-chemical parameters of the nine industrial facilities analyzed (three facilities for each type of bio-material). Moisture percentages were between 60 and 85%, except in a vegetable waste facility (VR3). The three raw materials analyzed were found, in at least one situation, in values higher than 80% humidity. This can lead to anaerobic conditions in the piles at the start of the composting process [30]. Therefore, conditioning operations should try to counteract these problems. Regarding the pH situation in raw materials, the behavior was restricted to the nature of the residue. Within the residues characterized in the study, the MSW had a typical slight acid character [31]. On the other hand, SS and green waste registered values above pH neutrality. On the contrary, in the conductivity of the bio-materials, with a clear difference, SS and MSW were more suitable as starting substrates than green wastes. The presence of organic matter was between 65 and 80% in all cases. Thus, all the residues contained a high organic load susceptible to composting. Despite the above, the form in which organic matter is present is highly variable and largely dependent on the origin of the waste [32]. Moreover, this is just what occurred in the present study. The C/N ratios did not reach the value of 10 in SS, nor the value of 15 in VR. In contrast, the ratio was optimal (20–30) for MSW. The respirometry of the bio-materials revealed the higher consumption of oxygen associated with MSW with respect to SS. In turn, it was found that the raw materials that reach the industrial facilities were substrates with a high content of fresh, non-stabilized organic matter. This fact was fully demonstrated through the result of the phytotoxicity of bio-materials. The full totality of the analyzed residues, presented values lower than 50% in the germination index, which is the threshold value of toxicity for this parameter [33].

**Table 2.** Properties of raw materials from industrial composting facilities *.

| Facility ** | M (%) | pH | EC (mS cm$^{-1}$) | OM (%) | C/N | AT$_4$ (gO$_2$ kg$^{-1}$ OM) | DRI (gO$_2$ kg$^{-1}$ OM h$^{-1}$) | GI (%) |
|---|---|---|---|---|---|---|---|---|
| **VR1** | 85.94 d | 8.10 ef | 16.78 c | 74.06 de | 12.63 d | 52.38 a | 1.07 ab | 7.70 ab |
| **VR2** | 66.62 b | 8.04 ef | 14.73 cd | 73.48 d | 13.30 d | 67.03 a | 1.70 b | 0.00 a |
| **VR3** | 29.27 a | 6.78 c | 13.53 d | 77.13 f | 10.61 c | 38.98 a | 0.46 a | 1.91 b |
| **MSW1** | 78.19 c | 5.16 b | 3.98 ab | 74.73 e | 28.47 f | 196.56 b | 3.01 c | 0.00 a |
| **MSW2** | 84.83 cd | 3.94 a | 2.65 a | 75.15 e | 28.05 f | 160.95 b | 2.73 c | 0.00 a |
| **MSW3** | 61.06 b | 7.22 d | 5.37 b | 71.73 c | 20.10 e | 49.65 a | 0.74 a | 0.00 a |
| **SS1** | 81.87 cd | 7.95 e | 3.02 a | 71.97 c | 9.29 b | 23.27 a | 0.62 a | 40.42 c |
| **SS2** | 66.37 b | 8.63 g | 2.56 a | 64.91 a | 9.50 ab | 57.83 a | 0.95 a | 18.70 b |
| **SS3** | 67.67 b | 8.40 fg | 3.06 a | 66.14 b | 6.44 a | 31.86 a | 0.61 a | 38.34 c |

* Data are mean values (n = 3), those with the same letter in the same column are not significantly different from each other (LSD, $p < 0.05$). Abbreviations: M: moisture; EC: electrical conductivity; OM: organic matter; C/N: carbon-nitrogen ratio; AT$_4$: dynamic accumulated respiration activity; DIR: dynamic respirometric index GI: germination index. All data are on a dry weight basis. ** Vegetal residue (VR); municipal solid waste (MSW); sewage sludge (SS).

### 3.2. Evolution of Soluble and Polymeric Fractions

In a composting process, the decomposition of polymeric residual matter occurs through microbial action. The microbiota consume nutrients in chemical forms solubilized and simple. If this is not the case, the nutrients must be biotransformed into metabolizable compounds. The exoenzymes secreted by the microorganisms produce the release of the basic sources of nutrients. This action allows the growth of the composting microbiome. Therefore, during composting there must be considerable consumption of the soluble

and polymeric fractions of the waste material. In the present study, both fractions were evaluated at the beginning and at the end of the process (Figure 1). This quantification allowed the observation of the degradation that these fractions suffered during composting (Figure 2). According to the results obtained in the polymeric fraction, the fibers content of the raw materials ranged between 30 and 40% of the dry weight (Figure 1a). As expected, and also reported earlier by others [34], the main fraction thereof, consisted of cellulose (10–20%). The fiber content was relatively constant among all the samples, regardless of the raw material. In contrast, the soluble fraction was highly dependent on it (Figure 1c). The SOC represented around 2.5% of the dry weight in MSW, but barely reached 0.2% in SS. On the other hand, in SS the majority of the soluble fraction was clearly the soluble protein. This result can be explained by the high content of nitrogen fraction contained in these residues due to their origin [35]. Nitric and ammonia fractions did not represent a considerable soluble fraction either in the SS or in MSW studied.

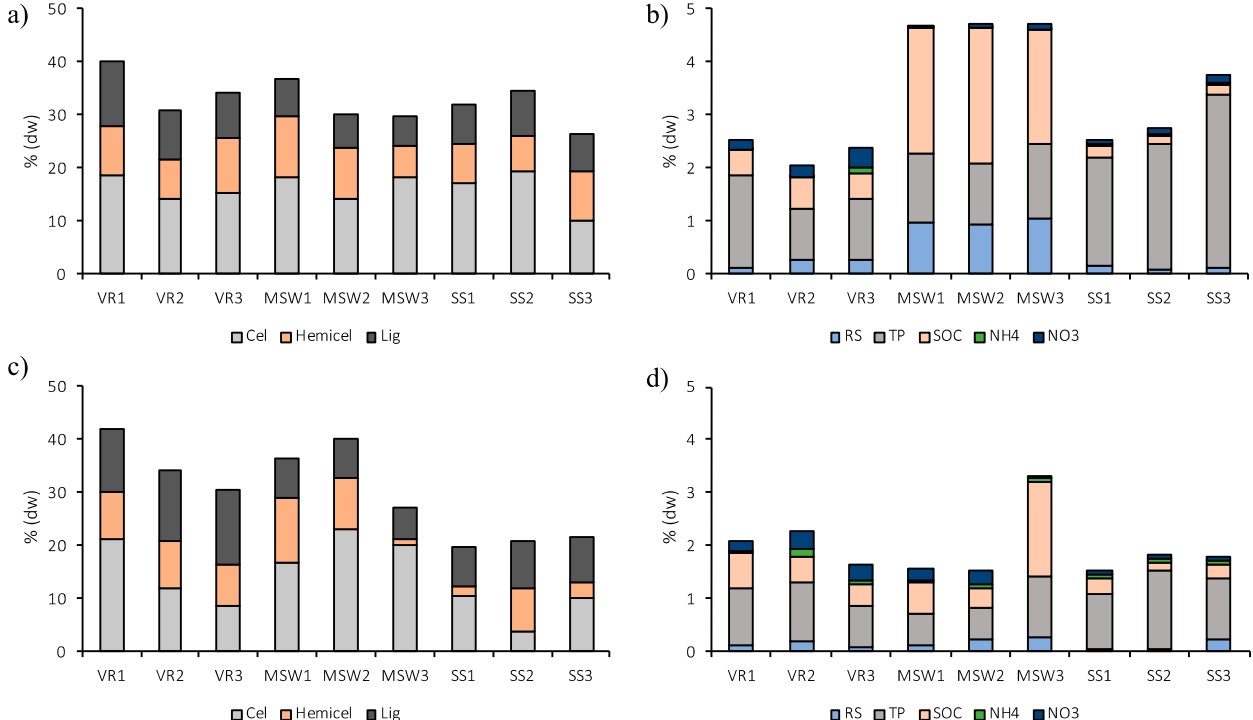

**Figure 1.** (**a**) Fiber fractions in raw materials; (**b**) soluble fractions in raw materials; (**c**) fiber fractions in compost; (**d**) soluble fractions in compost. Abbreviations: cellulose (Cel), hemicellulose (Hemicel), lignin (Lig), reducing sugars (RS), total proteins (TP), soluble organic carbon (SOC), N-NH$_4^+$ (NH4), N-NO$_3^-$ (NO3).

Throughout the composting process, the bio-materials undergo a series of biotransformations. These changes are detectable by analyzing the variation in the soluble and polymeric fractions [25]. Therefore, in this study both fractions were also quantified in the compost produced (Figure 1b,d) despite the fact that in industrial scaling processes, waste can often not be biotransformed as correctly as in ideal composting. The variable content of these fractions in composts revealed situations in which biodegradations were more intense. Figure 1b,d showed that the degradation of organic matter was clearly detected through the evolution of the soluble fraction. Zhang et al. [36] also previously described this behavior. The results of the fiber fractions require long periods of active biodegradation to be effectively mineralized. The recalcitrance of these polymeric structures causes a challenge to microbial degradation [37].

Despite the above, during composting it is common to find quantifiable degradative processes in all fractions of organic matter. That is precisely what was intended to be determined in this study. For this, the degradation percentages reached by the different

fractions were evaluated and are shown in Figure 2. The data were highly revealing to analyze the performance of the facilities. Degradation of the holocellulose fraction (hemicellulose + cellulose) was detected in up to seven of the nine facilities studied, with situations where this fraction was consumed by up to 50% (SS1 and SS2). In fact, these results differed greatly depending on the raw material to be composted. Thus, in MSW there was hardly a significant degradation of holocellulose while in SS it was intense. On the other hand, in neither of these two residual materials a consumption of the most recalcitrant fraction of the fibers (lignin) was reported. Regarding the soluble fractions, a considerable depletion of nutrients was detected in both residues. In fact, it was higher than that detected in green waste facilities. Although the use of this fraction by the microbiota produced variable results depending on the composted bio-materials. In this way, it was possible to corroborate a greater decrease in SOC levels in MSW, compared with that registered in SS. Even, this decrease far exceeded 70% in two MSW facilities (MSW1 and MSW2). On the contrary, according to the results of the SS facilities, in this bio-material, the consumption of the soluble fraction was mainly attributed to the reduction in the nitrogen fraction of the organic matter. The reduction in this nitrogen fraction, as a whole, is widely described in composting [38].

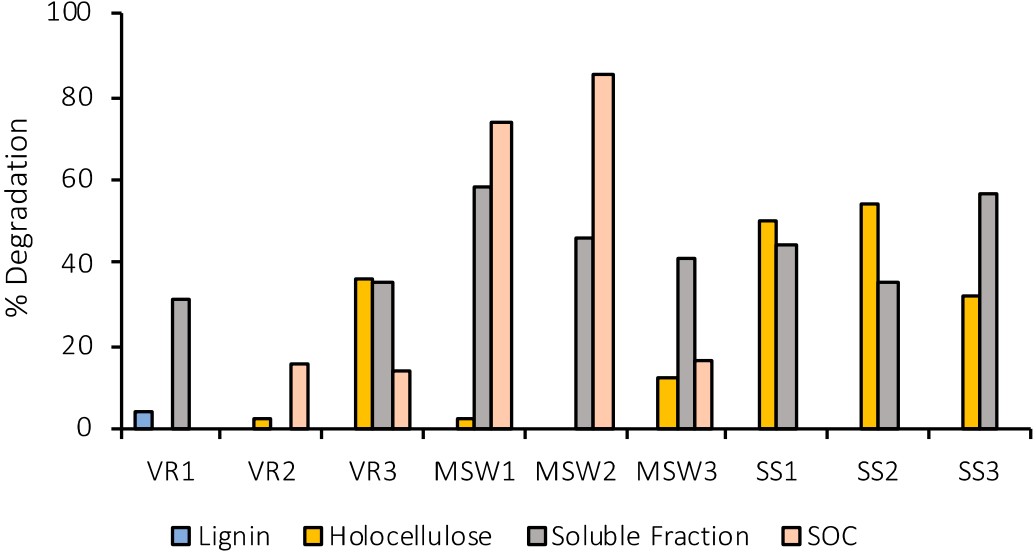

**Figure 2.** Percentage of degradation in organic fractions during industrial composting processes. Abbreviations: vegetal residue (VR); municipal solid waste (MSW); sewage sludge (SS); holocellulose (cellulose + hemicellulose); soluble fraction (include reducing sugars, total proteins, $N-NO_3^-$ and $N-NH_4^+$); Soluble Organic Carbon (SOC).

### 3.3. Fecal Contamination in Composts

The study of fecal contamination in the present study tracked four representative groups: coliforms (total, fecal and *Escherichia coli*), fecal enterococci, sulfite-reducing clostridia and *Salmonella* spp. The first three indicator groups were quantified in their levels in composts (Figure 3a–d), while for *Salmonella* spp., only the presence was searched. In addition, the recorded levels of enterococci and *E. coli* were compared with those established by current European legislation [39]. This legislation indicates that composts to be used as fertilizers cannot contain levels higher than 1000 cfu $g^{-1}$ in *E. coli* or enterococci (but not necessary in both groups) and must have an absence of *Salmonella* spp. in 25 g of compost. According to the results obtained, only one facility exceeded these microbial limits and contained *Salmonella* spp. in the compost, MSW2. The rest of the facilities fulfilled the legislation. Even so, four facilities exceeded the enterococci content (VR2, MSW2, MSW3 and SS3). However, they did not exceed the *E. coli* and *Salmonella* spp. limits, so they complied with the requirements. These results showed that the fecal enterococci group was more resistant to the conditions prevailing during composting than the *E. coli* and

*Salmonella* spp. groups. This behavior was also reported in composting by [40]. The justification for this resistance lies in the very nature of the group of enterococci. This microbial group, unlike the other two, is Gram-positive and has a slightly higher thermotolerance. Sulfite-reducing clostridia are not delimited in the European legislation. In spite of this, the quantification of this group is of great interest because it is a good indicator of fecal contamination, and complementary to those commonly used [41]. The counts obtained in this strict anaerobic group were found to be closely linked to the nature of the composted waste. Thus, those bio-materials where anaerobic situations are created, contained a higher amount of sulfite-reducing clostridia (SS > MSW > VR). In short, according to our results, the composting processes carried out in industrial facilities were able to generate adequate sanitization conditions to produce fertilizers that satisfy the current legislation; with the exception of the MSW2 treatment facility, where minimum thermal sanitization conditions were not maintained. The compost generated in the latter facility can only be disposed of in a controlled landfill.

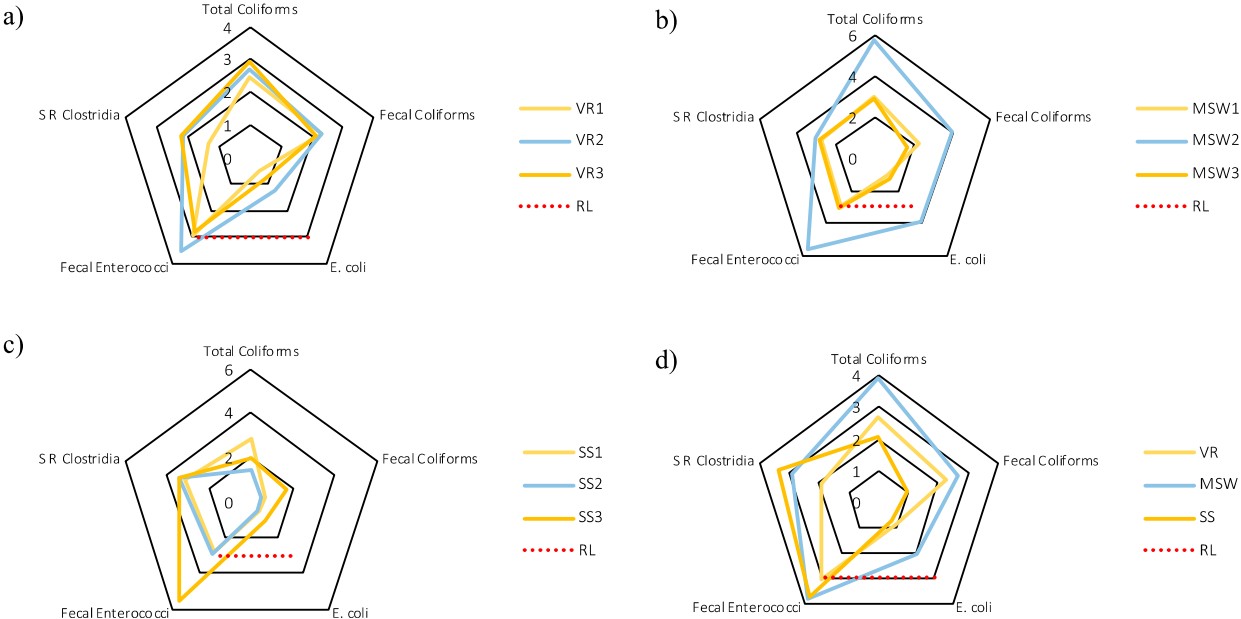

**Figure 3.** Fecal contamination levels in industrial facilities (Log (cfu g$^{-1}$ compost)). (**a**) Vegetal residue (VR) facilities, (**b**) municipal solid waste (MSW) facilities, (**c**) sewage sludge (SS) facilities, (**d**) all the industrial facilities analyzed per raw materials. Abbreviations: SR Clostridia: sulfite-reducing clostridia; R.L: regulation law according to [39].

### 3.4. Composts Characterization and Discriminant and Correlations Analysis

The characterization of the final composts produced was decisive in order to know in detail which processes resulted in adequately matured products. Results of the control parameters, together with the maturity indexes, are shown in Table 3. The reduction in the moisture content of the waste generates a severe reduction in microbial survival, with the consequent thermal decrease in the piles. Thus, on occasions of waste saturation, the dehydration of the material can cause a false sensation of waste stabilization [42]. MSW1 and MSW2 were placed under this premise. Regarding the final pH values, neither the sewage sludge nor the MSW contained a pH that could hinder plant growth. In fact, the most unfavorable values of this parameter were obtained in the green wastes (VR1 and VR3). In fact, this result is usual in the case of green wastes of horticultural origin [43]. In the same way, the electrical conductivity values found do not represent a hindrance to crops, except for MSW3 and the VR facilities. These control parameters significantly affect the development of composting. Therefore, they have an impact on the degree of mineralization reached during the process [44]. According to the results obtained, the degradation of organic matter in industrial composting was intense. The

values were between 26% (SS2) and 63% (VR2 and MSW3). Precisely, this interval marked the differentiation between the lowest and highest phytotoxicity detected among the samples. Therefore, the monitoring of organic matter was relevant to know the degree of maturity of the bio-materials. Similarly, the C/N ratio informs about the availability of unconsumed nutrients in the piles [45]. In the present study, this parameter obtained its maximum value in the most phytotoxic compost (MSW3) and the minimum in the most mature composts (SS1 and SS2). However, to know in detail the state of maturity and stabilization, it is best to use specific parameters for this purpose. According to the specialized literature on this subject, the most accurate measurements are those that include a respirometry and biological test [46]. This is precisely what was chosen in this study. The respirometric indices $AT_4$ and DRI presented minimum values in composts of the industrial processes of SS. In addition, the GI results established a separation between the processes that generated phototoxic (<50%), moderately phytotoxic (50–80%) and non-phytotoxic (80–100%) materials. As an overall result, only two industrial treatment facilities, both SS, were able to generate phytotoxic-free composts. This result highlights the hard work ahead for the improvement of waste treatment facilities at the industrial level.

**Table 3.** Properties of final compost from industrial composting facilities *.

| Facility ** | M (%) | pH | EC (mS cm$^{-1}$) | OM (%) | C/N | $AT_4$ (gO$_2$ kg$^{-1}$ OM) | DRI (gO$_2$ kg$^{-1}$ OM h$^{-1}$) | GI (%) |
|---|---|---|---|---|---|---|---|---|
| VR1 | 41.05 g | 9.18 f | 8.48 c | 48.43 c | 11.73 bc | 23.20 b | 0.47 cd | 45.79 bc |
| VR2 | 20.46 c | 8.08 d | 17.36 e | 63.30 f | 14.09 d | 30.05 b | 0.44 cb | 2.66 a |
| VR3 | 24.72 d | 9.68 g | 9.97 d | 39.64 b | 10.91 b | 25.72 b | 0.38 bc | 46.43 bc |
| MSW1 | 11.31 b | 8.66 e | 4.97 b | 53.91 e | 11.79 c | 34.23 c | 0.47 cd | 32.73 b |
| MSW2 | 5.66 a | 7.50 b | 5.58 b | 38.05 b | 15.63 e | 75.78 d | 0.73 d | 45.31 bc |
| MSW3 | 50.95 h | 6.00 a | 10.29 d | 63.65 f | 22.44 f | 30.23 b | 0.50 cd | 0.00 a |
| SS1 | 30.80 e | 7.72 c | 4.67 b | 47.19 c | 8.92 a | 1.32 a | 0.04 a | 91.08 d |
| SS2 | 32.33 f | 8.26 d | 2.72 a | 26.09 a | 9.63 a | 9.04 a | 0.20 acb | 99.80 d |
| SS3 | 32.39 f | 8.52 e | 5.52 b | 50.18 d | 14.21 d | 7.99 a | 0.09 ab | 52.68 c |

* Data are mean values (n = 3), those with the same letter in the same column are not significantly different from each other (LSD, $p < 0.05$). Abbreviations: M: moisture; EC: electrical conductivity; OM: organic matter; C/N: carbon-nitrogen ratio; $AT_4$: dynamic accumulated respiration activity; DIR: dynamic respirometric index GI: germination index. All data are on a dry weight basis. ** Vegetal residue (VR); municipal solid waste (MSW); sewage sludge (SS).

In this study, maturity indexes, physico-chemical parameters and microbiological contamination were evaluated in the composts produced. This allowed the performance of a statistical study that included a discriminant and correlation analysis. The discriminant analysis loading plot of control parameters, fecal contamination indicators and degradation percentages of the organic fractions from municipal solid waste (MSW), sewage sludge (SS) and vegetal residue (VR) is represented in Figure 4. Data are grouped in three classes: I, MSW; II, VR; III, SS. This analysis was performed including the parameters gathered in Table 3 and Figures 2 and 3. Two discriminant functions were obtained that explained 100% of the variation. The first discriminant function accounts for 80.33% of the variation and separated the samples into three groups in function of the raw material. The second function explained 19.67% of the variation and separated the green wastes of both urban wastes evaluated in the work. In both functions, the separation was mainly based on respirometry indexes, the fecal coliform group and holocellulose degradation. This indicates that both the origin of the waste to be treated and the sanitation and stabilization operations during composting were fundamental to generate composts of agronomic interest.

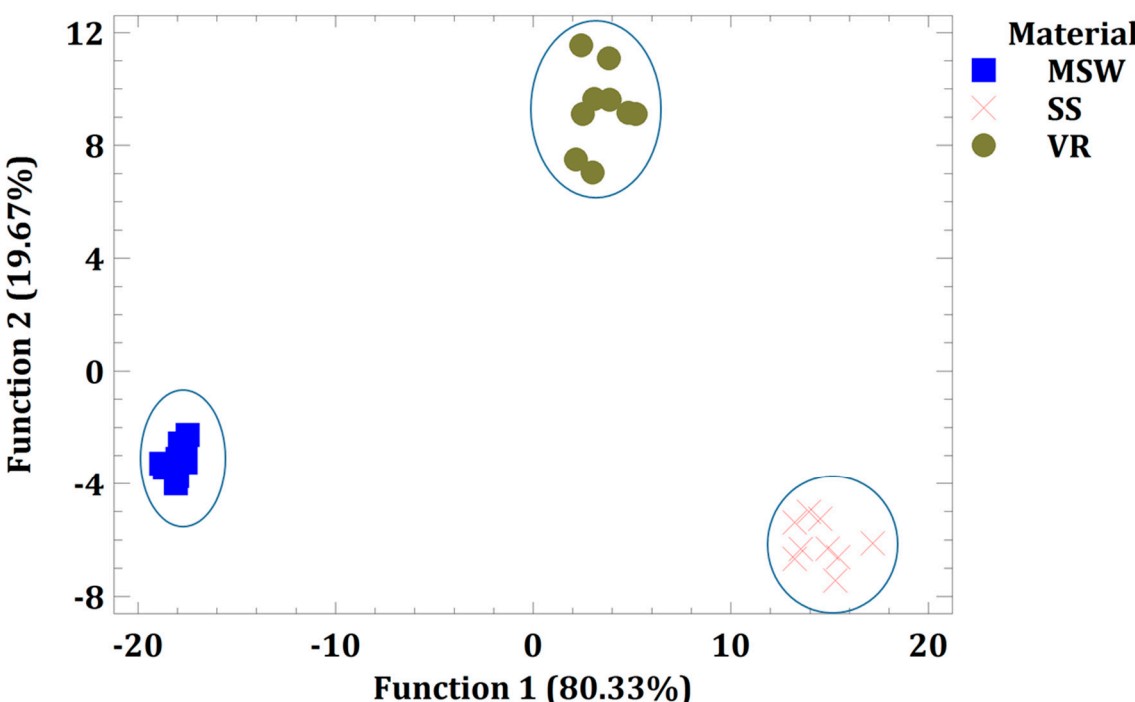

**Figure 4.** Discriminant analysis from municipal solid waste (MSW), sewage sludge (SS) and vegetal residue (VR).

Spearman's correlation analysis is shown in Table 4. This analysis provided a deeper understanding of the factors that affected the sanitization and stabilization of the biomaterials. As expected, the proliferation of fecal contamination indicator groups in composts was a clear symptom of the presence of fresh organic matter; therefore, of detectable phytotoxicity in GI or by respirometry (RDI and $AT_4$). Although very few articles have evaluated this relationship, there is literature that corroborates the results obtained [47]. On the contrary, when the consumption of the holocellulose fraction was increased, products that promoted plant growth were generated ($R^2 = 0.56$). This value indicated that the consumption of nutrients present in the residues was not only intense, but also served to produce stabilized composts. In short, although parameters of different nature were used, the results showed that, despite the intrinsic difficulties of industrial-scale composting to ensure optimal composting conditions, it is possible to predict and manage the process to correct the deficiencies of the starting raw materials.

**Table 4.** Spearman's correlation matrix between maturity indexes, physico-chemical parameters and microbiological contamination of industrial composts (n = 9).

| | %Degr S F | S R Clostridia | %Degr SOC | Fecal Enterococci | *E. coli* | Fecal Coliform | Total Coliforms | AT$_4$ | DRI | GI | C/N | OM | pH |
|---|---|---|---|---|---|---|---|---|---|---|---|---|---|
| **%Degr Holocellulose** | X | 0.42 | −0.53 | −0.42 | −0.41 | −0.76 | −0.43 | −0.70 | −0.62 | 0.56 | −0.61 | X | X |
| **%Degr S F** | | 0.48 | X | X | X | X | X | X | X | X | X | X | X |
| **S R Clostridia** | | | X | X | X | −0.48 | X | X | −0.56 | 0.47 | X | X | X |
| **%Degr SOC** | | | | X | 0.58 | 0.58 | 0.71 | 0.77 | 0.51 | −0.47 | 0.44 | X | X |
| **Fecal Enterococci** | | | | | 0.57 | X | X | X | X | X | 0.61 | X | X |
| ***E. coli*** | | | | | | 0.55 | 0.65 | 0.60 | 0.40 | −0.50 | 0.66 | X | −0.44 |
| **Fecal Coliforms** | | | | | | | 0.52 | 0.77 | 0.64 | −0.40 | 0.47 | X | X |
| **Total Coliforms** | | | | | | | | 0.56 | X | −0.41 | X | X | X |
| **AT$_4$** | | | | | | | | | 0.83 | −0.59 | 0.60 | X | X |
| **DRI** | | | | | | | | | | −0.56 | 0.48 | X | X |
| **GI** | | | | | | | | | | | −0.71 | −0.73 | X |
| **C/N** | | | | | | | | | | | | 0.50 | −0.45 |
| **OM** | | | | | | | | | | | | | X |

Abbreviations: %Degr Holocelullose: % degradation of cellulose + hemicellulose; %Degr S F: % degradation of soluble fraction (reducing sugars, total proteins, N-NO$_3^-$, N-NH$_4^+$); %Degr SOC: % degradation of soluble organic carbon; S R Clostridia: sulfate-reducing clostridia; AT$_4$: dynamic accumulated respiration activity, DRI: dynamic respirometric index; GI: germination index, C/N: carbon-nitrogen ratio, OM: organic matter. Blue color represents positive correlation and yellow color negative correlation. X indicates statistically non-significant values in the Spearman's correlation.

## 4. Conclusions

Frequently, urban organic waste treatment facilities are unable to generate bioproducts with sufficient quality to allow an agronomic use to the composts. According to the results obtained in the present study, most of these facilities are capable of ensuring adequate sanitation. Thus, they generate materials that do not constitute a risk of disease transmission, in accordance with current legislation. However, as reflected by the compost quality parameters used, proper sanitization was not a sufficient condition to generate mature and stable products. Only by enabling degradative conditions of the organic matter fractions contained in the waste, was a stabilization of the bio-materials achieved; therefore, a loss of phytotoxic compounds. The facilities that treated MSW had greater difficulties than those that treated SS. Therefore, more emphasis should be placed on the duration of the process, since the control parameters of these raw materials are suitable for composting. The use of the germination index and respirometric parameters facilitated the monitoring of the industrial process. In future research on industrial composting it may be advisable to introduce microbiological indicators. These parameters, together with those used in this study, may reveal the true operating strategy under industrial conditions, which ensures the production of high-quality compost.

**Author Contributions:** A.B.S.-C.: Investigation, Methodology, Formal analysis, Writing—original draft. J.A.L.-G.: Conceptualization, Formal analysis, Data curation, Investigation, Writing—review and editing. M.M.J.: Conceptualization, Investigation, Resources. F.S.-E.: Conceptualization, Investigation, Methodology, Resources. M.J.E.-G.: Investigation, Methodology. M.J.L.: Conceptualization, Supervision, Writing—review and editing, Data curation, Formal analysis. All authors have read and agreed to the published version of the manuscript.

**Funding:** This research was funded by the Ministry of Economy and Competitiveness, Spain, through project AGL2015-64512R.

**Institutional Review Board Statement:** Not applicable.

**Informed Consent Statement:** Not applicable.

**Conflicts of Interest:** The authors declare no conflict of interest. The funders had no role in the design of the study; in the collection, analyses, or interpretation of data; in the writing of the manuscript, or in the decision to publish the results.

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
