# Peer review of "Compost Quality and Sanitation on Industrial Scale Composting of Municipal Solid Waste and Sewage Sludge"

_applsci, doi:10.3390/app11167525_

Round 1

Reviewer 1 Report

The manuscript “ Compost quality and sanitation on industrial scale composting of municipal solid waste and sewage sludge” is interesting and within the scope of the journal but some minor changes should be addressed:

  1. Please try to short the figure titles by inserting the information in the manuscript text.
  2. I recommend to update the state-of-the-art by discussing briefly about potential unconventional techniques for sewage sludge derived products obtaining (please see https://doi.org/10.37358/RC.20.10.8361). It will be also interesting to discuss about the migration of metals from waste into environment in order to highlight the ecological risk (please see https://doi.org/10.37358/RC.19.12.7725).
  3. Please mention in the materials and methods section from how many points were collected the samples.
  4. Please use subscript (e.g. line 112).
  5. I recommend to use equation at line 140.
  6. Please try to extend the conclusion section by bringing the own opinions or future direction.

Author Response

The comments are very much appreciated, we thank the referee. Please see the attachment.

Reviewer 2 Report

This manuscrpit presents the results of study conducted in order to evaluate the processes of sanitation and the quality of the compost produced form municipal solid waste and sewage sludge.

Prior acceptance, this manuscript should be carefully read, checked and technically correct.

Doing things, authors should be specially focused on following:

  • all units should be written in the same form
  • chemical formulas should be carefully check and rewrite
  • the information about the results of this study should be incorporated in the section Conclusions

Author Response

The corrections are very much appreciated, we thank the referee. We have corrected and marked in red everything indicated by the reviewer. Please see the attachment

Round 2

Reviewer 2 Report

The manuscript can be accpeted.